# Rapid development of fast and flexible environmental models: The Mobius framework v1.0

Magnus D. Norling[1], Leah A. Jackson-Blake[2], José-Luis G. Calidonio[1], James E. Sample[2]

[1]Norwegian Institute for Water Research, 0349 Oslo, Norway

[2]Norwegian Institute for Water Research, 4879 Grimstad, Norway

*Correspondence to:* Magnus D. Norling (magnus.norling@niva.no)

**Abstract.** The Mobius model building system is a new open source framework for building fast and flexible environmental models. Mobius makes it possible for researchers with limited programming experience to build performant models with potentially complicated structures. Mobius models can be easily interacted with through the MobiView graphical user interface

and through the Python programming language. Mobius was initially developed to support catchment scale hydrology and water quality modelling, but can be used to represent any system of hierarchically structured ordinary differential equations, such as population dynamics or toxicological models. Here, we demonstrate how Mobius can be used to quickly prototype several different model structures for a dissolved organic carbon catchment model and use built-in auto-calibration and statistical uncertainty analysis tools to help decide on the best model structures. Overall, we hope the modular model building

platform offered by Mobius will provide a step forward for environmental modelling, providing an alternative to the "one size fits all" modelling paradigm. By making it easier to explore a broader range of model structures and parameterisations, users are encouraged to build more appropriate models, and in turn this improves process understanding and allows for more robust modelling in support of decision making.

## 1 Introduction

Environmental models are increasingly used both to formalise the current state of scientific knowledge and to support policy and practical decision making. There is therefore a strong need for robust models. Present-day system knowledge, new data sources (for instance from remote sensing) and the practical experience of many modellers suggest that model structure and complexity should be tailored to the system of interest based on (i) the research or policy question of interest, (ii) the response characteristics of the system, and (iii) the availability of observational data. When developing a model application, a

compromise therefore needs to be reached between the realism with which natural processes are represented as a set of equations, and whether those processes are important for the specific objectives of the modelling exercise, whether it is computationally feasible to represent those processes and whether data is available to evaluate the hypotheses put forward by the model (Clark et al., 2015). Within catchment hydrology and water quality modelling, for example, the dominant processes that dictate the system response to external drivers vary according to the scale of interest, with different processes revealing

themselves to be the main driver as scale increases from plot to hillslope to catchment (e.g. Sivapalan et al., 2003). Appropriate process representation may also vary from catchment to catchment (Kavetski & Fenicia, 2011; Wagener et al., 2007). Despite this, many popular water quality models provide limited flexibility for customising model structure. For example, spatially semi-distributed models typically allow users to define their own hydrological response units (HRUs), but the process representation within each HRU is essentially fixed. It is common for users to make implicit changes to the model structure,

for example by setting parameter values to zero or ±infinity, but this approach is both inflexible and opaque. Inflexible model structures are problematic when attempting to generalise complex models developed in areas with detailed monitoring data to less well-studied regions with sparse data. Overfitting is a serious problem for many environmental models and several authors have expressed concerns regarding testability or falsifiability of model simulations (see e.g. Kirchner, 2006). The predominance of inflexible model structures has lead to a vast number of models, often developed for a specific location or to

answer a specific question and then (perhaps inappropriately) transferred elsewhere (e.g. hydrology models are considered in Beven, 2012). To tackle this issue, calls have been made for open source community models (e.g. Mooij et al., 2010; Weiler & Beven, 2015), and/or so-called "models of everywhere", where the idea is to move from generic models that are customised to particular locations, for example through appropriate parameterisation, to models that are specific to particular places (Blair et al., 2019).


Modular (or flexible) modelling systems try to address these issues by providing a unified computational environment within which models can be developed, and therefore offer an attractive and increasingly popular alternative to the "one size fits all" paradigm. A well-designed framework makes it possible to quickly explore alternative model structures, and to explicitly customise existing models for specific applications. Model evaluation and comparison then become significantly easier, as all

model variants can use the same input and output data formats and share components of the same code base where appropriate. The user can therefore be certain that differences in output are due to intended structural changes, and not to implementation differences (e.g. in data pre-processing, secondary model components, or numerical solving schemes), meaning more significant scientific insight can be gained when comparing alternative process representations (Mooij et al., 2010). Modular frameworks also make it possible for users to extend or combine previously developed models – for example by quickly

developing multiple variants of a water quality model, all underpinned by the same hydrology module.

Although modelling frameworks have many advantages compared to traditional (fixed structure) approaches, they are difficult to implement in practice as building a flexible, generalizable framework requires considerably more programming expertise than building a static model. Many modellers have skills in interpreted languages such as Python or R, which are reasonably

well-suited to rapid prototyping of alternative model structures. However, exploring uncertainty in model predictions, or formally comparing model structures, typically requires methods such as Bayesian MCMC, which involves running each model thousands or even millions of times. This can be a big limitation for models coded in interpreted languages, which are

comparatively slow. One solution is to build models using compiled languages like C++ or Fortran, but many modellers lack the time or inclination to develop the necessary programming skills.


This paper presents a new modelling framework, Mobius, which allows flexible and fast model building by researchers with a relatively basic level of programming. Mobius models meet modern demands for computational speed, and allow for the complexity of process representation to be varied depending on progressing system knowledge, research question or scale. Several hydrological model building frameworks exist to date, such as FUSE (Clark et al., 2008), SUPERFLEX (Fenicia et al., 2011), FARM (Euser et al., 2013) and SUMMA (Clark et al., 2015). These all allow predefined components to be connected in user-specified ways to create a model, with a focus on catchment hydrology. The framework presented here takes these existing approaches further, allowing the user to define any component/process. It is therefore, to our knowledge, one of the first frameworks to be fully generalizable: although initially developed to support catchment-scale hydrology and water quality modelling, it can be used to represent any system of hierarchically structured ordinary differential equations (ODEs), such as population dynamics or toxicological models. Moreover, the Mobius framework does more of the heavy lifting of organizing the program structure than what we find in other frameworks. The user can create models with a complex structure without having to organize that structure explicitly using programming architecture. A range of popular hydrology and water quality models have already been implemented in Mobius, for instance the INCA-family of models (Futter et al., 2007, 2014; Jackson-Blake et al., 2016; Wade et al., 2002; Whitehead et al., 1998) and the Simply models (Jackson-Blake et al., 2017), and these are available to use either as standalone models or as starting points for further development and customisation.

In the remainder of the paper, we first describe the core of the Mobius framework (Section 2 ). We then describe tools for interacting with Mobius models, including the MobiView application (Section 3.1.1), a user-friendly graphical user interface (GUI) compatible with all Mobius models, and the Mobius Python wrapper (Section 3.1.2), which provides Python bindings to core Mobius functionality and incorporates many powerful optimisation and uncertainty estimation tools from the Python ecosystem. We then demonstrate the utility of Mobius by developing a new illustrative dissolved organic carbon catchment model, including rapid development of a variety of potential model structures, and using the tools available through the Python wrapper to choose an appropriate structure for model application in a Norwegian catchment (Section 3.2). We also include a model run speed benchmarking test, to demonstrate that performance is only marginally compromised by the increase in flexibility (Section 3.3). We finish by discussing the current scope and limitations of the framework, as well as future plans (Section 4 ).

## 2    Overview of Mobius

Mobius is a general framework for building models consisting of ordinary differential equations (ODEs) and equations evaluated only once per time step (discrete time step equations). Only limited programming knowledge is needed to build or

modify Mobius models. When building a Mobius model, the user specifies what state variables are in the model and what equations govern these. Equations can depend on the values of input time series, such as meteorological forcings, and be tuned using parameters. The core Mobius framework is built using highly optimised C++ code, but users can create new models or adapt existing ones by adding new model equations, parameters, inputs and dependencies between these, without having a detailed understanding of C++.

When programming models without a framework one must typically do a significant amount of work when adding in new processes or parameters. This may involve putting the mathematical equations in the right place in the model evaluation code or even restructuring the model evaluation code, recording the value of the equations at each time step for later exporting, packing various values into structures for use inside ODE integrators (something that can be daunting to inexperienced programmers), updating parameter file formats and parsers, and updating the code that exports the result time series to the desired final format. Any user interface or plotting code for that is designed to visualising the new process will typically also need updating. The Mobius framework automatically takes care of these things, allowing the researcher to focus only on the mathematical formulation of the processes. Even with this flexibility, run speeds are comparable to custom-coded C++ models, and considerably faster than models written in languages such as Python or R (see Section 3.3).

The framework exposes two application programming interfaces (APIs): the model builder API and the model interaction API. The model builder API allows for specification of models and modules. Finished models can be interacted with through the model interaction API. The following is an overarching description. More detail is given in "Documentation\model_builder_documentation.pdf" in the Mobius repository. The core implementation is described in the "Documentation\framework_implementation_documentation.pdf" and in the source code itself (see Section 5 ).

## 2.1    The model builder API

The model builder API allows a model creator to register model entities by name (a string), and associate metadata with them. Each registration procedure returns a handle that is used to refer to that entity. Model entities include parameters, input time series and equations (state variables).

The code that is used to evaluate each equation is provided in-line in the model building code. Each equation can use the (present or past) value of other entities that have been registered by referring to them by their handles. Each equation must return a single value per evaluation. In the simplest case, each equation is evaluated once per time step, resulting in a time series. An equation that is registered as an ODE is instead integrated over the timestep, typically using smaller sub-timesteps (the size of the sub-timesteps can depend on convergence criteria).

In more complicated setups, parameters and inputs can index over one or more index sets. Index sets can be viewed as "response units" in a loose way, meaning for instance a river reach or sub-catchment in a catchment model, a size class or

species in a biological population model, or a grain size and density class for microplastic particles. For example, in a catchment model some processes may have different parametrisations depending on land use class or sub-catchment. The land use index set could then contain the indexes "Forest" and "Agricultural" to allow for separate evaluation of the common processes for these two land use types.

The equations are automatically distributed over auto-generated arrangements of batches. Batches are groups of equations that 130   are evaluated for each (tuple of) index(es) in some collection of index sets:

- If an equation uses the (current-timestep) output of another, it will be evaluated after it.
- An equation will be indexed by all the index sets that index the entities it makes use of the value of.
- Equations are grouped into batches that index over the same index sets if they can be ordered consecutively according to the above criteria. In special cases, batch grouping is also determined by what ODE solver is used.

For instance, if an equation depends on a parameter that has a separate value per "Landscape unit", then that equation will also have a separate value per "Landscape unit". Furthermore, special types of equations and value access syntax allows an equation to aggregate the value of another equation over an index set or access values of entities from a different tuple of indexes to the one being currently considered. In this way, one can for instance let flow from different parts of the landscape go into a single river.

As an example, say that one wants to describe the input of water to the soil $q_{in}$, which in this example is different per land use class. Let $i$ be the time step (daily in this case, but the framework allows for configurable time step sizes) and let $u$ be a land use class. Let $p_{rain}$ be the amount of rain per day, and $t_{air}$ be the air temperature (these could be input time series – in this short example we omit how we compute snow fall). Let $DDF_{melt}$ be a degree-day factor for snow melt (a parameter) and let $d_{snow}$ be the computed snow depth. Then a simple equation describing the water input to soil could be

$$q_{in,i}^u = p_{rain,i} + \min\left(d_{snow,i-1}^u, DDF_{melt}^u t_{air,i}\right)$$

This system can be added to a Mobius model as follows (code for computing snow depth and registration of units is omitted).

```
auto LandscapeUnits = RegisterIndexSet(Model, "Landscape units");
auto SnowParams     = RegisterParameterGroup(Model, "Snow parameters", LandscapeUnits);
auto DDFmelt        = RegisterParameterDouble(Model, SnowParams, "Degree-day snow melt factor",
MmPerDegCPerDay, 2.74, 1.6, 6.0);
auto RainFall       = RegisterInput(Model, "Rain fall", MmPerDay);
auto AirTemperature = RegisterInput(Model, "Air temperature", DegC);

auto WaterInputToSoil = RegisterEquation(Model, "Water input to soil", MmPerDay);
auto SnowDepth        = RegisterEquation(Model, "Snow depth (water equivalents)", Mm);

EQUATION(Model, WaterInputToSoil,
        return INPUT(RainFall) + min(LAST_RESULT(SnowDepth), PARAMETER(DDFmelt)*INPUT(AirTemperature));
)
```

Since the "Degree-day snow melt factor" belongs to the "Snow parameters" parameter group, which is set to index over the "Landscape units" index set, the equation "Water input to soil" will (when the model is later run) be evaluated per land use
class since it uses the value of this parameter. If one also makes it so that "Rain fall" is indexed over the "Sub-catchment" index set, the equation is evaluated per pair of (landscape unit, sub-catchment) indexes. Setting the index set dependencies of inputs can even be done during model interaction stage, such as through an input file. Note that "Degree-day snow melt factor" is given a unit and default, min and max values. These are not used in the model run, but they are metadata that can e.g. be displayed in a UI to guide the model user.

What indexes each index set contains can be specified during the later model interaction step, where they can e.g. be loaded from a parameter file or set by a GUI. Land use and river connectivity structure in a catchment model, for example, can thus be specified without changing or recompiling the model.

ODE equations are organized into batches by the framework depending on the selected integrator algorithm. At present, one Runge-Kutta 4 ODE integrator based on the DASCRU algorithm (Wambecq, 1978) is bundled with Mobius, and there are
wrappers for the Boost Odeint solvers (Ahnert et al., 2011). Other solvers can be made accessible in Mobius without having to modify the core framework code.

When a model is run, all state variables (equations) are recorded each time step, resulting in a time series corresponding to each of them. When equations are evaluated for multiple indexes, a time series is produced for every index combination. In this way you may for instance get a separate "Soil water volume" time series for each sub-catchment and land use class in a
hydrology model. This exhaustive recording facilitates introspection into model processes, for instance in the MobiView user interface (Section 3.1.1), which can be used during all stages of model development. This way a model can be built iteratively, adding one process at a time, and assessing how it performs.

The modular system in Mobius allows various modules to be combined. A module is simply a procedure that registers entities with a model and provides the associated equation code. A module can access entities that were registered in other modules.
This is done by obtaining the handle to the other entity by loading it from its string name. One benefit of the fact that the framework organizes the main evaluation loops of the model is that ODE equations from a module can be solved in the same ODE integrator batch as the ODE equations from another module without any effort by the model creator. In this way one module can extend the ODE systems of another, creating a larger coupled set of equations, or even override individual equations from the other module without creating a separate version of the source code of that other module.

**2.2    The model interaction API**

Any model created using the model builder API can be compiled to a dynamically linked library (.dll on Windows, .so on Linux) or can be included into another C++ project. The model interaction API (which is exported through the library) can be

used to create programs that use the model for specific purposes. We have created a wrapper for the API in the Python programming language (Section 3.1.2). It is possible to call these functions from any language with a C foreign function interface. Some of the things one can do with this API are:

- Create a "dataset" object that can be structured for a concrete setup of the model with given parameter values and input time series.
- Set specific index set structures in the dataset. In catchment models this allows for setting up the river structure and land use classes.
- Run the model one or more times with a given dataset. One can also make thread safe (and fast) copies of the dataset to run the model multiple times in parallel.
- Read the resulting time series of any model equation (for a given index tuple corresponding to this equation's index set dependencies).
- Read the full structure of the model and accompanying metadata that was registered in the model building process. One can e.g. extract a list of the names of all the parameters or equations, their units, descriptions, index set dependencies, etc.

Model entities are interacted with by referring to the string name that they were registered with. Mobius supports a custom text format for parameter and input files that is tailor made to be convenient to edit by hand. A json format for parameter and input files is also supported, which could for instance be used for serialisation and web communication. The API also makes it easy to add support for new file or data formats.

The MobiView user interface (Section 3.1.1) is an example of a program built using this API. Other options are making command line applications, R modules, (for instance using Rcpp, see Eddelbuettel & François, 2011), wrappers to other languages, or making complex setups such as running the same model for multiple climate scenarios, sensitivity analysis, ensemble runs of multiple models, or autocalibration (more on this in Section 3.1.2). The API can also be used to couple Mobius models to models that were not created in Mobius.

## 3    Demonstration of Mobius

### 3.1    Tools for convenient interaction with Mobius models

Compiled Mobius models can be interacted with in two user-friendly ways, using a GUI or Python.

### 3.1.1    MobiView GUI

The MobiView GUI can interact with any model that is compiled using the standard Mobius .dll interface. It is ideal for model users and developers to quickly explore Mobius model parameters, equations and inputs, and carry out manual calibration.

The GUI displays a structured organisation of parameters, associated descriptions and recommended ranges. It is easy to find parameters using a name search. The workflow for manual calibration is convenient: the user can update a parameter value, then click a button (or keyboard hotkey) to re-run the model and immediately see the results in the plot view (Figure 1).


There are various plot modes and ways to customise the plotting to be able to analyse the model result time series. For instance, the user can switch between daily values and monthly and yearly aggregations. There are also residual plots, residual distribution histograms and quantile-quantile plots for analysing the performance of a result time series compared to an observed time series. MobiView computes several different goodness-of-fit statistics (including for example bias, mean

absolute error, mean squared error and Nash-Sutcliffe efficiency; Nash & Sutcliffe, 1970). Any observation time series can be loaded for use in calibration. Other features include visualisation of branched river structures (for hydrology models), ability to export results to csv formats, customisation of plot visual layout and export of plots to image or pdf formats.

MobiView is developed using the Ultimate++ framework and the ScatterCtrl package (ultimatepp.org).

**3.1.2    Python wrapper and integration with model auto-calibration and uncertainty analysis packages**

Users can interact with compiled Mobius models using a Python wrapper. Python is a high-level programming language well suited to rapid development and prototyping, as well as being more accessible to domain scientists than compiled languages such as FORTRAN or C++. Python also offers a wide range of additional packages, including tools for model optimisation, calibration and uncertainty analysis. ODE-based models implemented in "pure" Python often suffer from poor performance.

By implementing the model using Mobius and communicating with it via the Python wrapper, users can therefore benefit from both the performance of C++ and the flexibility and modules available in Python. Although the wrapper adds a small computational overhead, it generally offers excellent performance, because for most ODE-based models with realistic levels of complexity, the main performance bottleneck will be running the model itself and not communicating through the Python interface.


The wrapper makes it easy for users to modify input time series and parameter values, run the model and extract time series of results. This makes it convenient to script many types of sensitivity and uncertainty analysis setups that are reusable across different models. Functions are provided for plotting and visualising outputs, and for calculating a range of commonly used goodness-of-fit statistics. It is also straightforward to connect Mobius models to other tools in the Python ecosystem and to

parallelise multiple model runs across many processes or cores. For example, auto-calibration can be implemented by defining a Python function to update parameter values, run the model and return an appropriate summary of the results (such as the sum of squared errors). This "loss function" can then be minimised using any of the tools available via Python.

The current Python wrapper provides access to generic functions to aid model auto-calibration and uncertainty estimation. Key dependencies are the Python packages LMFit (Newville et al., 2014) and emcee (Foreman-Mackey et al., 2013). LMFit offers a consistent interface to a range of optimisers (Levenberg-Marquardt, Nelder-Mead etc.), as well as providing a 'Parameters' class that allows users to define plausible parameter ranges (e.g. "priors" in the context of a Bayesian analysis) and to choose which model parameters should be varied and which fixed. Auto-calibration using LMFit is typically fast and most optimisers return estimates of confidence intervals for the fitted parameters. It therefore provides an excellent starting point for further investigation. For more complex models with potentially multi-modal likelihoods/posterior distributions, or for users wishing to explore parameter-related uncertainty in more detail (e.g. by explicitly specifying a likelihood function), emcee provides a state-of-the-art Markov chain Monte Carlo (MCMC) algorithm based on the affine-invariant ensemble sampler (Goodman and Weare, 2010). Emcee's ensemble sampler supports various methods for parallelisation and is well-suited to exploring the complicated and inhomogeneous posterior distributions characteristic of many ODE-based environmental models. Although more computationally intensive than optimisation, sampling using MCMC provides much richer information describing the (Bayesian) posterior probability of the model's parameters, given the calibration dataset and the underlying assumptions. The Python wrapper includes functions for visualising MCMC chains and creating "corner plots" of the posterior distribution, which provide valuable diagnostic information that can be used to inform iterative refinement of the model structure within the core Mobius framework. We give examples of how this can be used in Section 3.2.4.

A convenient approach to interacting with Mobius models via the Python wrapper is to use Jupyter Notebooks (Kluyver et al., 2016), which provide an effective platform for well-documented, shareable and reproducible modelling workflows. The Mobius GitHub repository (see Section 5 ) provides example code illustrating how to interact with models via the Python wrapper, including auto-calibration of the nutrient model SimplyP (See the "PythonWrapper\SimplyP\simplyp_calibration.ipynb" file in the repository.

## 3.2     Rapid model development – a case study

We now demonstrate how Mobius can be used to easily build a variety of model structures, and then how the tools made available through the Python wrapper can be used to decide on an appropriate model structure for a particular study area, given the observed data available. To illustrate this, we will develop some simple example alternative model structures to simulate daily river dissolved organic carbon (DOC) concentrations in a small upland catchment in Norway. Stream DOC concentrations have been rising in recent decades in many regions around the world due to a combination of recovery from acidification and climate change (Monteith et al., 2007, de Wit et al., 2016), and model predictions of potential future changes are of interest in terms of drinking water quality, carbon cycling and climate feedbacks.

### 3.2.1 Case study site and data for model selection

The study site is a small stream and associated catchment in southeast Norway, one of the main inlets to the lake Langtjern (510–750 m.a.s.l; 60.371 N, 9.727 E). The catchment has an area of 0.8 km$^2$ and land cover is 80% pine forest on thin mineral soils and 20% bog on deeper peat. Mean annual temperature, precipitation and discharge (1986-2015) are 2.5°C, 901 mm and 650 mm, respectively.

Water discharge and DOC observations were used for model selection. Discharge is difficult to simulate in this catchment, due to a combination of short water residence times and a flashy hydrology, and uncertainty in the observed discharge (which until 2014 was based on a water balance for the lake, see de Wit et al., 2018) is high. Since 1986, stream water grab samples have been collected weekly to monthly and analysed for total organic carbon (TOC) (see de Wit et al., 2014, for details of sampling methods and chemical analysis). In this catchment, TOC is essentially equivalent to DOC. Starting in August 2014 there is also daily soil temperature data (at 15 and 20 cm depths).

### 3.2.2 General model set up

The modelling aim is to reproduce long-term daily in-stream DOC concentrations rather than a detailed carbon balance. All DOC model versions are built on a common hydrology module, SimplyQ, which was developed for SimplyP (Jackson-Blake et al., 2017), excluding the deeper soil flow path. In brief, water and associated DOC may be transported from the land to the stream via 'quick' flow (infiltration and saturation excess overland flow and deeper bypass flow) and/or shallow soil water flow, which is somewhat slower. DOC fluxes to the stream are therefore simply via soil water flow, given by $Q_s[DOC]_s$, and via quick flow, as $Q_{quick}[DOC]_s$, where $Q_s$ is soil water flow, $Q_{quick}$ the quick flow, and $[DOC]_s$ is the soil water DOC concentration. Soil water and quick flow vary through time as a function of precipitation, evapotranspiration, soil moisture levels and runoff to the stream. The factors which control the variation through time of $[DOC]_s$ are investigated through the model selection process described below. As there are no upstream inputs in our study area, the mixing of these different water sources gives the in-stream DOC concentration.

Models were run for the period 1986-2016 using input meteorological data (air temperature and precipitation) from a local weather station operated by met.no, the Norwegian Meteorological Institute.

### 3.2.3 Carbon model structures

The model development process starts with a statistical exploration of the observed data and knowledge of the literature, and these together are used to generate a list of potential processes to include, and different possible formulations for a given process. These are then translated into a range of model structures. In this example, a strong correlation was found between observed stream DOC concentration and modelled soil temperature, but there are questions as to the appropriate representation

300    of this process, and longer-term processes and hydrological effects such as snowmelt dilution may also be important. To this end, six model structures were developed (the DOC-related model parameters are defined in Table 1):

1. Simple linear relationship between soil water DOC concentration and soil temperature. The linear relationship describes an empirical relationship between equilibrium soil water DOC concentration and soil temperature, $T_{soil}$:

$$[DOC]_s = [DOC]_{s,base} + k_{T,1}T_{soil}$$

   Soil temperature is computed based on air temperature using a simplified version of the Rankinen soil temperature model (Rankinen et al., 2004). We assume that the DOC concentration reaches equilibrium instantly.

2. More complex relationship between soil water DOC concentration and soil temperature than structure 1, a second-degree polynomial:

$$[DOC]_s = [DOC]_{s,base} + (k_{T,1} + k_{T,2}T_{soil})T_{soil}$$

3. The observed DOC time series shows a long-term trend which is not explained by soil temperature, but which other studies have suggested is due to recovery from soil water acidification (Futter & de Wit, 2008; Monteith et al., 2007). To test the importance of this process, we use yearly means of measured stream $SO_4^{2-}$ concentration as a proxy for soil water acidification and add a linear dependence of DOC concentration on $SO_4^{2-}$ concentration:

$$[DOC]_s = [DOC]_{s,base} + (k_{T,1} + k_{T,2}T_{soil})T_{soil} - k_{SO4}[SO_4^{2-}]$$

   Note that in an operational model, this would need replacing with $SO_4^{2-}$ concentration in deposition (which should be well correlated with stream water $SO_4^{2-}$ concentration) to allow for future predictions.

4. There are visible short-term decreases in the stream DOC concentration during snow melt, likely due to source-exhaustion and dilution. In this structure, we attempt to simulate this by introducing a separate parameter for the snow melt DOC concentration:

$$(\text{quick DOC flux}) = Q_{quick,melt}[DOC]_{melt} + Q_{quick,rain}[DOC]_s$$

5. Starting from Structure 3, replace the soil temperature model by the Lindström model (Lindström et al., 2002) to see if the effect of the choice of soil temperature model is important.

6. Instead of assuming instant equilibration of soil water DOC concentration, add equilibration as a delayed process. We add a state variable $[DOC]_{s,eq}$ which obeys the same equation as $[DOC]_s$ in the formulation from structure 3. We then let DOC mass in the soil move toward the point where equilibrium is satisfied:

$$[DOC]_{s,eq} = [DOC]_{s,base} + (k_{T,1} + k_{T,2}T_{soil})T_{soil} - k_{SO4}[SO_4^{2-}]$$

$$\frac{dDOC_s}{dt} = c_{eq}([DOC]_{s.eq} - [DOC]_s) - [DOC]_s Q_s$$

$$[DOC]_s = DOC_s/V_s$$

   where $V_s$ is the modelled soil water volume and $c_{eq}$ is the equlibration speed factor.

Going from one structure to the next typically involves just a few lines of code (see code and data for this experiment in the Mobius repository. Application files for compiling models, input data files and parameter files are in the "Applications\SimplyC_paper" subfolder, while module files containing the definitions of each structure (inputs, parameters and equations) are in the "Modules\Alternate_versions_of_simplyC" subfolder).

### 3.2.4 Model comparison and selection of the most appropriate structure

The model structures were calibrated using data from the period 1986-2003. The calibrations were then evaluated on data from the period 2004-2016. All auto-calibrations were performed using the implementation of the Nelder-Mead algorithm in the Python LMFIT package, described in Section 3.1.2. We auto-calibrated the hydrology module separately first and fixed the hydrology parameters for all subsequent calibrations of the DOC-related parameters. It is possible to calibrate for hydrology and DOC at the same time, but in this particular experiment we decided to simplify the parameter space to only those parameters relating to DOC, to allow more targeted model selection. The two soil temperature modules used were also calibrated just once each against the more limited soil temperature data available.

For each model structure in order:

1. We manually calibrated the DOC-related parameters. If applicable, manual calibration used the parameter values from the auto-calibration of the previous structure as a starting point. The manual calibration targeted the Nash-Sutcliffe coefficient as the goodness-of-fit statistic.

2. Auto-calibration was run using the manual calibration as a starting point. The auto-calibration algorithm uses a least-squares fitness measure. In terms of finding optimal parameters, this is equivalent to optimising for the Nash-Sutcliffe coefficient.

Auto-calibration of models written in the Mobius framework is fast due to the fast run speeds of the models. The longest time needed to auto-calibrate any of these structures was 189 seconds (time depended on the number of parameters calibrated). See more on benchmarking in Section 3.3. This aids with quick evaluation of new model structures.

Goodness-of-fit statistics from the automatic calibrations of each of the six model structures are given in Table 2, together with the number of calibrated DOC-related parameters. A plot of the observed vs. modelled stream DOC concentration using the auto-calibrated parameter sets for structures 1, 3 and 5 is shown in Figure 2.

Visually, the results are good overall, but all structures fail to capture high DOC concentrations during some summers. The improvement of fit from structure 1 to 2 is obvious (Table 2), as structure 2 allows for a more flexible relationship between soil temperature and soil water DOC concentration and this relationship is a strong determining factor for stream DOC

concentration in this catchment. The improvement from structure 2 to 3 is not as large, but structure 3 does capture long-term trends a little better – in structure 2 there is a long-term trend in the residuals that disappears in structure 3 (data not shown).

Adding snow melt dilution in structure 4 does not give a significant improvement of fit. This is possibly because the snow model used is simple and not constrained by observed snow levels, so that the timing of the snow melt may be off. Moreover, snow melt happens during a short time span, and so will not register as strongly when just calibrating for DOC concentrations. If one also calibrated for total DOC fluxes, it would be more prominent due to the high water flow during snow melt, which would be something to explore further for an operational model. Changing the soil temperature model in structure 5 obtains a better fit for soil temperature, but the stream DOC fit was relatively unchanged, probably because differences in modelled soil temperature could be compensated for by variations in the parameters that determine soil DOC response to soil temperature. Structure 6 captures melt dilution better, but it creates too much noise in the signal the rest of the year. Calibrating for goodness-of-fit tends to adjust the $c_{eq}$ parameter to be very high so that equilibration is almost instant (i.e. the model is close to equivalent to earlier formulations).

Out of the 6 structures, model 5 performed marginally best during validation, but had two additional parameters compared to Structure 3. Overall, structure 3 seems to be the most appropriate given the observed data, offering the best compromise between model performance (particularly during validation) and complexity. More work would be needed to arrive at an operational model, and more formal model selection process using e.g. a Bayesian approach would also be possible (e.g. Marshall et al., 2005). However, hopefully this exercise serves to illustrate the relative ease with which model development can be carried out and alternative structures quickly explored.

To explore how well-constrained parameters in Structure 3 are by the observed data, and also to explore any parameter covariance, we then used the emcee algorithm (see Section 3.1.2) to generate a sample of the posterior distribution of structure 3 and associated marginal posteriors of the parameters. The model run interval was the same as the earlier calibration interval. The sampler was run with 100 chains for 1000 steps, each with a burn-in of 100 steps, and showed good convergence. A heteroscedastic Gaussian error structure was used, where the standard deviation of the likelihood at each point in time is assumed to be equal to a Bayesian error parameter ($err_{\mathrm{DOC}}$) multiplied by the simulated value at that point. A corner plot of the results (Figure 3) shows that the parameters are well constrained by the observed data - which is desirable in a model - and gives an idea of the probable range of each parameter, represented as the 95% "credible interval" on each marginal histogram. Clear covariance between $[\mathrm{DOC}]_{s,base}$ and $k_{SO4}$ is also apparent.

## 3.3    Benchmarking of model run speeds

For benchmarking, we created a hard-coded test model in C++ (i.e. the model code was written without using a framework) and a mathematically equivalent model in Mobius. The model was a simple hydrological model (SimplyQ; Jackson-Blake et al., 2017), and we verified that the two implementations produced the same results up to numerical error. The hard-coded model had a straightforward implementation and was not excessively optimised using advanced techniques such as single instruction multiple data (SIMD) or optimisation of cache locality, but we assume that this is not commonly done by most researchers. A hard-coded version of the model was also produced in Python. Results of the benchmarking show that Mobius models have a slight performance loss compared to hard-coded C++ models but run several orders of magnitude faster than hard-coded Python models (Table 3). Code used in these experiments can be found in the "Evaluation" subfolder of the Mobius repository.

Note that we only report the ratio of how fast the other implementations run compared to the Mobius implementation, since that is what shows the comparative advantage or disadvantage of using Mobius when it comes to model run speed. This ratio is relatively stable across the machines we tried (for instance an Intel Core i5-6600K CPU 3.50 GHz, and an Intel Core i5-8350U CPU 1.70GHz).

## 4    Discussion and outlook

Mobius aims to be a framework for rapid development of hydrological and biogeochemical models and other models based on ODE and discrete time step equations. Model development should be fast and flexible, and models should run quickly. The aim is for Mobius to be a virtual environmental laboratory for researchers to test their hypotheses about natural processes using quantifiable data.

Each component of the Mobius framework targets a different user group:
- Practitioners with little or no programming experience can use MobiView to manually calibrate and apply existing models that have been built by other users.
- Researchers with basic programming skills can use the Python wrapper to perform sophisticated model auto-calibration and uncertainty assessments, make predictions under uncertainty and consider whether model process representations are adequate.
- Researchers and developers with more advanced programming skills can use all components of the framework to iteratively calibrate, evaluate and refine process representations and/or make ensemble simulations from a range of model structures.

We find that Mobius satisfies its aims for a large range of models. More complicated ones than the one described in Section 3.2 can be found in the Mobius repository (see Section 5 ). The automatic model structure generation and the optimization

and visualization tools allow the user to quickly formulate models and test them without having to do technical programming.

With models that have more complicated process descriptions (many equations) or rely on many different compartments or index sets, it can take some more training to use Mobius correctly. This is because it takes training to create dependencies between index sets, parameters and equations that will create the intended structure. It may also take some experimenting with the model design to understand what index sets a model should use. To help with this, we have included many helpful debugging facilities such as detailed error messages, printouts of the generated structure and other statistics. There is also a detailed user manual and many existing examples.

Since the code that computes the state variables should be factored out as separate code pieces that can be evaluated in order, we find that Mobius is not that suitable for models where simultaneous computation of several state variables is needed (unless they can be described as a system of ODEs, which is well supported). This can happen when using certain linear algebra operations or when doing computations where several iterations are needed per time step in order to e.g. compute an equilibrium of many chemical compounds. It is possible to build such models in Mobius (the value of other state variables can be set from a single code piece), but the framework is not as helpful with this type of model.

There is a performance drawback of evaluating equation code through lambdas that are stored in an array, which is how Mobius stores the equation code, as that requires calling this code through function pointers. This is difficult for the compiler to optimize and can cause cache misses. A way to improve this would be to instead have a code generator that generates the model run structure as program code based on the model specification code, which is then compiled separately. The drawback of doing it that way is that it forces the framework creators to maintain their own parser for the model specification code, and it would make the model structure non-malleable after compilation.

There are a few other technical limitations in the current implementation:

- Strong two-way links (such as two-way fluxes) between different instances of the same equation batch are not well supported, though workarounds are possible in simple cases. For instance, it would currently be difficult to build grid-based models with an ODE-based two-way diffusion of quantities between neighbouring cells (assuming the number of cells is not fixed by the model). This is related to the next limitation.

- ODE equations in the same batch can be solved as one coupled ODE system for each index in the index set of the batch. But it is currently not possible to solve a coupled system consisting of multiple indexes at the same time. Instead, they must be solved as separate systems. This limitation only applies when using the automatic distribution of equations over indexes. If one instead manually codes every instance of the equations, this limitation is circumvented, but the flexibility of using the automatic indexing system is removed.

- Coupling between different Mobius modules works well, and the model interaction API can be used to couple Mobius models to models not build in Mobius. However, currently it is not possible to have per-timestep interaction between Mobius models and external models. Either the entire Mobius model must be run first to then use its outputs as inputs to the other model, or the other way around.

460 We hope to remove some of these limitations in the future. Indeed, Mobius is under active development, and priorities for the near future include expanding the range of available pre-built models (porting existing models from elsewhere into Mobius and developing new ones), developing interfaces for the R and Julia programming languages and development of a statistical model comparison framework to aid in model selection.

We are keen to build an open source community of users interested in modular open source model development, and to that 465 end we also plan on creating a user forum, carrying out training workshops, and to continue development of on-line documentation, tutorials and tools for easy interaction with models.

Overall, the Mobius framework combines cutting-edge computational speed with sophisticated model inspection and evaluation tools. This permits comprehensive model assessment – crucially including consideration of structural uncertainty – without compromising performance. The framework is freely available under a GNU Lesser General Public License (v3.0) 470 and we hope that by making it easier to explore a broader range of model structures and parameterisations, users will be encouraged to build better and more appropriate models. We believe this will in turn improve both process-understanding and practical decision making.

## 5   Code and data availability

The most up to date version of Mobius can be found at https://github.com/NIVANorge/Mobius. An archived version 475 (27.01.2020) corresponding to the work described in this paper is available on Zenodo doi:10:5281/zenodo.368211. Mobius is distributed with the GNU Lesser General Public License (v3.0). Mobius models can be compiled to work on most platforms, and MobiView works on Windows and Linux. Pre-compiled binaries of MobiView and selected Mobius models are available for 64-bit Windows (instructions on how to download are given on the Mobius GitHub front page).

## Supplementary material

480 User manuals and documentation for Mobius and MobiView are available in the "Documentation" subfolder of the Mobius repository (see Section 5 Code and data availability). See also the README file in the repository.

**Author contribution**

MDN developed the Mobius framework and MobiView, co-developed the Python integration of Mobius, implemented the DOC model examples and performed some of the experiments with them. LJB developed the Simply models and co-developed the DOC model example and performed some of the experiments with them. JES co-developed the Python integration of Mobius with the lmfit and emcee packages. JLG was involved in the design process of Mobius. MDN prepared the manuscript with contributions from all co-authors.

**Acknowledgements**

Mobius takes many of its design ideas from and supersedes the INCA Core Framework, developed by Dan Butterfield. The development of Mobius was partially funded by Nordforsk "Nordic eScience Globalisation Initiative (NeGI)" via the project "An open access, generic ePlatform for environmental model-building at the river-basin scale" (Machu-Picchu). We acknowledge the crucial role of Raoul M. Couture and Martyn N. Futter in getting that project started. Mobius development was also partially funded by the Norwegian Institute for Water Research (NIVA), and we acknowledge the very important support that Heleen de Wit and Thorjørn Larssen provided during development.

**Competing interests**

The authors declare that they have no conflicts of interest.

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

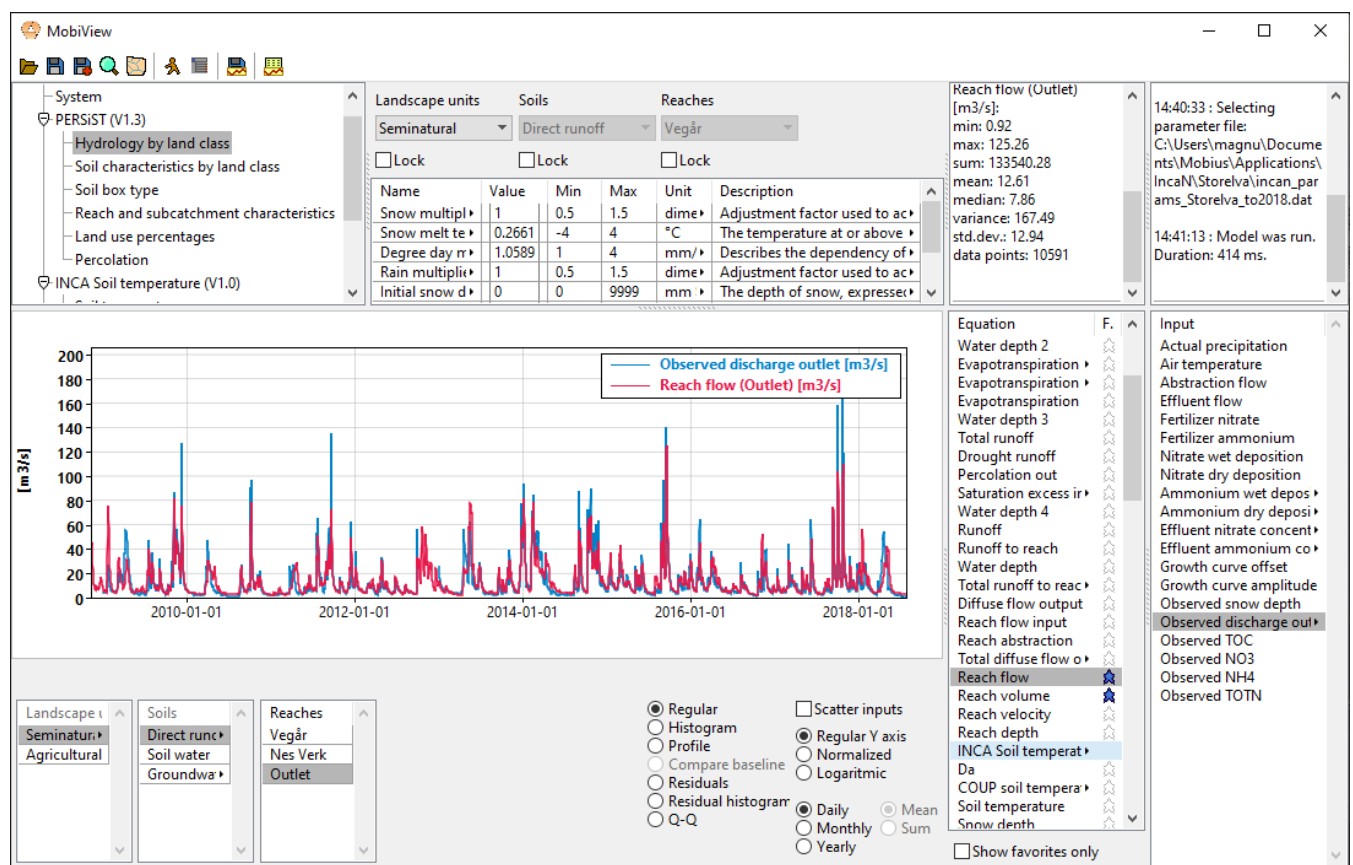

**Figure 1: The MobiView user interface, running INCA-N, a catchment nitrogen model. The image shows the module and parameter group structure of this model (top left), some editable parameters (top center), and a plot which shows a comparison between modelled and observed flow of water in the river. Any time series shown in the Equation and Input lists on the right can selected for plotting here.**

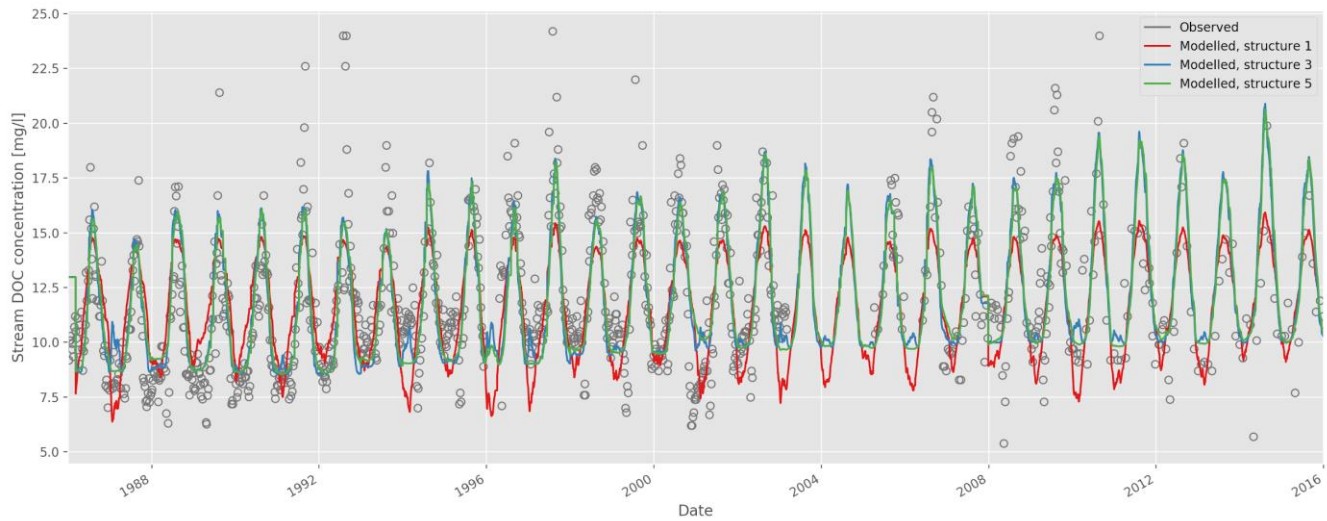

**Figure 2: Time series of observed and modelled stream dissolved organic carbon (DOC) concentration (model structures 1, 3 and 5; auto-calibrated parameters).**

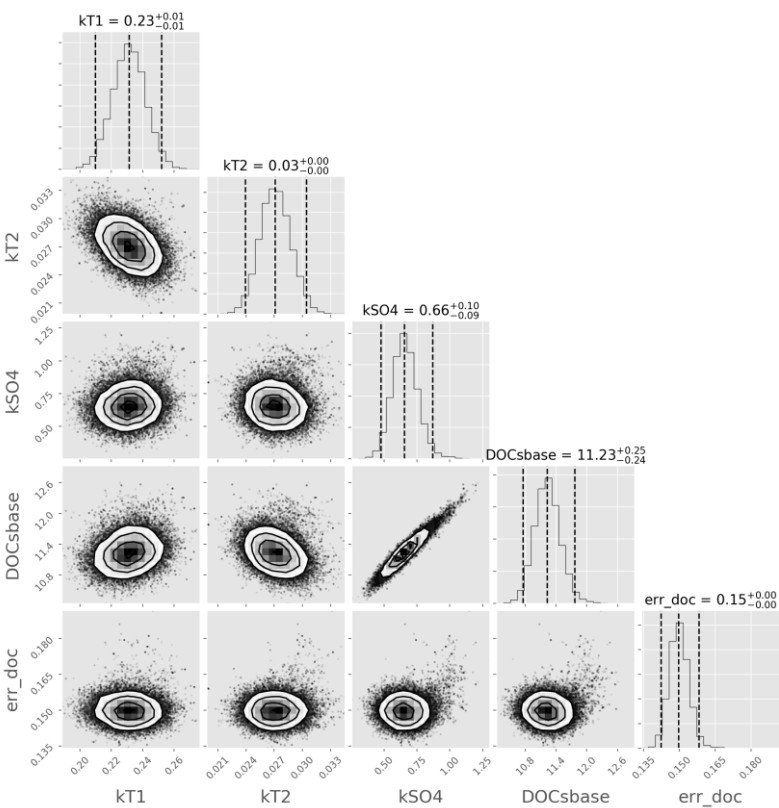

**Figure 3: Corner plot of the marginal posterior distributions of DOC-related parameters in model structure 3. Sampled using the emcee algorithm.**

| Parameter | Symbol | Model structure |
|---|---|---|
| Baseline soil water DOC concentration | $[DOC]_{s,base}$ | 1-6 |
| Soil temperature DOC concentration linear response coefficient | $k_{T,1}$ | 1-6 |
| Soil temperature DOC concentration second-order response coefficient | $k_{T,2}$ | 2-6 |
| Soil carbon solubility response to $SO_4^{2-}$ deposition | $k_{SO4}$ | 3-6 |
| Snow melt DOC concentration | $[DOC]_{melt}$ | 4 |
| Equilibration speed factor | $c_{eq}$ | 6 |

**Table 1: DOC-related parameters used in the various simple carbon model structures explored.**

| Model structure | Goodness of fit | | Number of parameters | |
|---|---|---|---|---|
| | Calibration (1986-2003) | Validation (2004-2016) | DOC | Soil T |
| 1 | 0.51 | 0.50 | 2 | 2 |
| 2 | 0.65 | 0.61 | 3 | 2 |
| 3 | 0.67 | 0.62 | 4 | 2 |
| 4 | 0.67 | 0.62 | 5 | 2 |
| 5 | 0.65 | 0.63 | 4 | 3 |
| 6 | 0.64 | 0.62 | 5 | 3 |

**Table 2: Goodness-of-fit (Nash-Sutcliffe coefficient) obtained for stream DOC concentration using the 6 model structures, and number of parameters relating to DOC processes and soil T (parameters based on well-constrained physically-measured quantities are excluded). Nash Sutcliffe values of 1 indicate a perfect fit; values of 0 suggest the model is no better a predictor than the mean**
**of the observations.**

| Hard-coded model language | Run speed ratio |
|---|---|
| **C++** | 0.5 - 0.7 |
| **Python** | approximately 200 |

**Table 3: Ratio of times taken for 1000 runs of the hard-coded model versions versus 1000 runs of the Mobius version. Testing was done on several common desktop machines and laptops, both under Linux and Windows.**