# Peer review of "Rapid development of fast and flexible environmental models: The Mobius framework v1.0"

_Geoscientific Model Development, 2020_

## Referee Comment (RC1) · Anonymous Referee #1 · 10 Jul 2020

As a model development paper, the contributions of the authors are valuable. The authors have provided various levels of development sophistication in their code: quick/preliminary model developments they using a gui, high-performance models using a C++ base, and a python wrapper for the middle ground. The contributions seem sufficient for a v1.0 paper and their commitment to open source paradigm is laudable.

My comments and questions for improvement are:

* As a reader interested in general ode models paragraph 280 is not clear. If the goal is to compare manual vs autocalibration of model parameters the starting points should be independent (perhaps random in feasible ranges?). Why do the authors use the

results of manual optimization to start the auto-calibration?

* If the authors foresee that the changes in code would break the experiments and files referenced in the paper they can include the commit number (or date) that preserves the experiments since the repository seems to be actively developed.

* The benchmark of runtime experiments can use a better description. Paragraph 335 mentions: "Results of the benchmarking show that Mobius models have a slight performance loss compared to hard-coded C++ models but run several 335 orders of magnitude faster than hard-coded Python models (Table 3)". If the source files for these benchmarks are also included in the github repository, please reference them in the paper.

* The authors should give a simple description of the hardware they are using to run tests. Just the manufacturer, number of cores, and frequency of the CPU is enough. This will ensure the timings have enough context.

* The timing experiments are average model evaluation runs. If possible( at least for the Mobius model using the python interface), It would be valuable to report optimization times in a separate table.

---

## Short Comment (SC1) · 20 Jul 2020

* The goal was not to compare manual to autocalibration, but to show one possible workflow when calibrating the model and iterating on model structures (i.e. use auto-calibration to "fine-tune" a manual calibration). We are not the authors of the autocalibration algorithms, and so the point is not to show off how good these are at reaching good solutions from random starting points. We are only showing one possible way these python libraries can be used with models built in Mobius.

* Future edits to the github repository should not break the experiments described in the paper. However, we also made an archived version of the repository corresponding

to this paper, which is linked to in section 5 (line 376).

* The source files for the benchmarking are referenced in the paper, line 335 "Code used in these experiments can be found in the "Evaluation" subfolder of the Mobius repository"

* Description of the hardware used in benchmarking can be provided in the revised manuscript. Note that we only report the ratio between the run times of different implementations. The run times themselves, are highly hardware dependent, while the ratios are fairly stable across several different machines we tried. Number of cores is not going to be that relevant since one model run is single threaded (while you can parallellize if you want to run the model many times such as in MCMC).

* I'm not sure what precisely is asked for in the last comment. Optimization times are highly variable, depending on algorithm, the model used, input data etc. These together determine how many model evaluations are needed to reach convergence, and are independent of the model implementation (as long as the implementations give identical results). The model implementation only determines how fast each model evaluation is, and so that is what we focused on in the benchmarking. Timing for optimization will scale accordingly. Nevertheless, we did provide the timing for one particular optimization run on line 287 (using one specific algorithm and model setup).
* * *

---

## Referee Comment (RC2) · Anonymous Referee #2 · 3 Nov 2020

General comments

This paper presents a framework to support the development of environmental models, referred to as the Mobius framework (v1.0). The aim is to allow scientists with potentially limited programming skills to develop component models within the framework, which can then be linked together. This is an important and timely contribution as new approaches to modelling are urgently required particularly as we head towards the need for integration of models. The framework is developed for hydrology and water quality analyses but should be applicable to other settings. The framework is also available as an open source tool with a link provided in the paper to the relevant github

repository. This is a well-written and accessible paper. My own major concern is how it is framed. It is written very much as a description of the approach rather than as a research paper. To be a fully-fledged research paper it would have: research questions and/ or a guiding hypothesis, consideration of the state of the art and gap identified, methodology and evaluation/reflections/discussion. These elements are largely missing. The paper would be much stronger being re-framed as more of a research contribution. I pick up on these points in my more specific comments below.

Specific comments

The introduction does a good job of motivating the research and I very much welcome the arguments presented in the paper. However, as mentioned above it is not framed as a research paper. It could be though with a bit of refactoring, for example, the paper claims things like flexibility and ease of use wrt novice programmers... these could be hypotheses that are evaluated through the research. The same argument applies to improvements over 'fixed models'. This is something that could also be evaluated. The second section provides an overview of Mobius. I found it quite hard to get to the crux of the design, and it is quite short and lacks any real depth. I am a computer scientist by training and I wanted to see things like an overarching software architecture and also an explanation of key design decisions with rationale. This is missing from me. It would be very hard fro example for other researchers to consider the text here and get anywhere near reproducing 'the approach'.

Section 3 is then a 'demonstration of Mobius' and this title says a lot about the way the paper is framed. To me, it should not be a demonstration of a given approach but rather should be an evaluation of how well the approach achieves its goals, with the evaluation being rigorous and scientific. Instead, it steps through the GUI (but curiously not in a visual way) and also the use of Python wrappers (the key to interoperability in their approach), but not in a way that allows the reader to see beyond the "what" to the "why" this is done (and other alternatives that could have been considered). This section also contains a case study – but again its stated purpose is to demonstrate not

to evaluate. It is also quite a small example and it is not clear how this would scale up to something more substantial. The section concludes with some benchmark figures, which are interesting, but it is not clear why performance is measured and nothing else is evaluated, when performance is not mentioned as a goal. Section 4 contains a discussion but to me this is way to narrow and specific and lacks a true element of reflection, e.g. what has worked, what has not worked, what are the strengths of the approach and weaknesses, and so on.

There is also so much more could be done in such a framework and these dimensions are not considered, e.g. running the model multiple times, perturbing parameters to carry out sensitivity analyses, running ensembles of models, looking at model coupling in a more sophisticated way, and so on. Finally, there is a lack of consideration of related work and yet there are a significant number of other frameworks in existence with similar goals.

In summary, I do think this is an interesting and potentially significant project but the paper needs significant revision to reach the stage where it can be published. In particular, it needs to be reframed as a research contribution in my view rather than a description of a particular approach.

---

## Author Comment (AC1) · 19 Nov 2020

We thank the reviewer for taking their time to do the review.

**As a reader interested in general ode models paragraph 280 is not clear. If the goal is to compare manual vs autocalibration of model parameters the starting points should be independent (perhaps random in feasible ranges?). Why do the authors use the results of manual optimization to start the auto-calibration?**

The goal was not to compare manual to autocalibration, but to show one possible work-flow when calibrating the model and iterating on model structures (i.e. use autocalibra-

tion to "fine-tune" a manual calibration). We are not the authors of the autocalibration algorithms, and so the point is not to show off how good these are at reaching good solutions from random starting points. We are only showing one possible way these python libraries can be used with models built in Mobius. Feasibility of autocalibration would also be highly dependent on how complex the model structure is. This could be a selling point for a given model (it is easy to auto-calibrate), but the model we present is just there to illustrate how one can use the framework, and this model itself is not the main point of the paper.

**If the authors foresee that the changes in code would break the experiments and files referenced in the paper they can include the commit number (or date) that preserves the experiments since the repository seems to be actively developed**

Future edits to the github repository should not break the experiments described in the paper. However, we also made an archived version of the repository corresponding to this paper, which is linked to in section 5 (line 376).

**The benchmark of runtime experiments can use a better description. Paragraph 335 mentions: "Results of the benchmarking show that Mobius models have a slight performance loss compared to hard-coded C++ models but run several orders of magnitude faster than hard-coded Python models (Table 3)". If the source files for these benchmarks are also included in the github repository, please reference them in the paper.**

**The authors should give a simple description of the hardware they are using to run tests. Just the manufacturer, number of cores, and frequency of the CPU is enough. This will ensure the timings have enough context.**

The source files for the benchmarking are referenced in the paper, line 335 "Code used in these experiments can be found in the "Evaluation" subfolder of the Mobius repository". Description of the hardware used in benchmarking can be provided in the revised manuscript. Note that we only report the ratio between the run times of

different implementations. The run times themselves, are highly hardware dependent, while the ratios are fairly stable across several different machines we tried. What we want to show is how fast a framework-implemented model runs compared to other implementation options.

The number of cores is not going to be that relevant since a single model run is single threaded (while you can parallellize if you want to run the model many times such as in MCMC).

**The timing experiments are average model evaluation runs. If possible (at least for the Mobius model using the python interface), It would be valuable to report optimization times in a separate table.**

I'm not sure what precisely is asked for here. Optimization times are highly variable, depending on algorithm, the model used, input data etc. These together determine how many model evaluations are needed to reach convergence. The number of evaluations to reach convergence are independent of the model implementation (as long as the implementations give identical results). The model implementation only determines how fast each model evaluation is, and so that is what we focused on in the benchmarking. In other words the time for the optimization is roughly N*t, where N is the number of evaluations and t is the time for a single evaluation (the total time can be reduced if the algorithm allows paralellisation, and t can be variable due to different convergence speed of ODE solvers depending on parameter values). The number N is not controlled by the model implementation, but by the optimization algorithm (and also the specific use case). Since the concern of this paper is the model implementation (framework), we focus on the single-run time in the benchmarking.

Nevertheless, we did provide the timing for one particular optimization run on line 287 (using one specific algorithm, model setup and machine) just to give a general idea that using autocalibration is feasible in fast-iteration model building.

[Figure]

2020.

---

## Author Comment (AC2) · 19 Nov 2020

**This paper presents a framework to support the development of environmental models, referred to as the Mobius framework (v1.0). The aim is to allow scientists with potentially limited programming skills to develop component models within the framework, which can then be linked together. This is an important and timely contribution as new approaches to modelling are urgently required particularly as we head towards the need for integration of models. The framework is developed for hydrology and water quality analyses but should be applicable to other settings. The framework is also available as an open source tool with a link**

[Figure]

**provided in the paper to the relevant github This is a well-written and accessible paper. My own major concern is how it is framed. It is written very much as a description of the approach rather than as a research paper. To be a fully-fledged research paper it would have: research questions and/ or a guiding hypothesis, consideration of the state of the art and gap identified, methodology and evaluation/reflections/discussion. These elements are largely missing. The paper would be much stronger being re-framed as more of a research contribution. I pick up on these points in my more specific comments below.**

We thank the reviewer for taking their time to do the review! The main criticism by the reviewer is that the paper is not framed as a research paper. Please note that the paper was submitted as a 'Model description paper', which is a form outlined by GMD, and allows for descriptions of models and modelling frameworks. See https://www. geoscientific-model-development.net/about/manuscript_types.html Substantially evaluating if using a (or this) framework vs not using one is productive would make this into a different kind of paper, but that was not our intended scope.

We will address some of the specific comments:

**The second section provides an overview of Mobius. I found it quite hard to get to the crux of the design, and it is quite short and lacks any real depth. I am a computer scientist by training and I wanted to see things like an overarching software architecture and also an explanation of key design decisions with rationale. This is missing from me. It would be very hard fro example for other researchers to consider the text here and get anywhere near reproducing 'the approach'.**

We can go into some more detail in the description of the software implementation details. Note that the form outlined by GMD says that implementation details should be included if they will substantially affect the output results. For models this is straightforward, but for frameworks what this implies is a little more up to interpretation. Note

also that the documentation included in the repository (and the source code itself) is an important part of the supplementary material to the paper, which does go into more detail. See especially https://github.com/NIVANorge/Mobius/blob/master/Documentation/model_builder_documentation.pdf for a detailed description of how one uses the framework to build models.

**Section 3 is then a 'demonstration of Mobius' and this title says a lot about the way the paper is framed. To me, it should not be a demonstration of a given approach but rather should be an evaluation of how well the approach achieves its goals, with the evaluation being rigorous and scientific. Instead, it steps through the GUI (but curiously not in a visual way) and also the use of Python wrappers (the key to interoperability in their approach), but not in a way that allows the reader to see beyond the "what" to the "why" this is done (and other alternatives that could have been considered). This section also contains a case study – but again its stated purpose is to demonstrate not to evaluate. It is also quite a small example and it is not clear how this would scale up to something more substantial. The section concludes with some benchmark figures, which are interesting, but it is not clear why performance is measured and nothing else is evaluated, when performance is not mentioned as a goal. Section 4 contains a discussion but to me this is way to narrow and specific and lacks a true element of reflection, e.g. what has worked, what has not worked, what are the strengths of the approach and weaknesses, and so on.**

There is a screen shot of the GUI, which shows a substantial part of its functionality. We can see if there is more we can do to make it more clear how it is used. (note that in the submission form of the manuscript all figures are at the end of the document. Presumably they will be inserted at appropriate places during editing). The GUI is not the main focus of the paper, but is described because of how it aids in making the framework easy to use both during development and for external users of the models.

The purpose of the demonstration section is to show that the framework allows you to

build, modify and evaluate models, with the goals that are stated earlier in the paper, among others:

1. it is very little work to modify models in order to evaluate different model structures.

2. It is easy to reuse existing modules (in this case a hydrology module)

3. Speed of model runs (performance) (this is definitely one of the important stated goals, see e.g. the paragraph around line 60)

4. Models can easily be run many times such as in auto-optimization and sensitivity analysis, which are important tools for evaluating model success. The goal of this is stated on line 60, and ties in to why it is important that the models run fast. We show that having auto-optimization and sensitiviy algoritms that can be used with any model built in the framework is useful because it allows you to very quickly apply them to any newly created model structures with little extra effort.

To see if the claims hold one can also inspect the source code used in the examples, which is given as a part of the supplementary material. Putting all the source code in the main paper would in our opinion be excessive.

We could make it more clear in the intro to section 3.2 that performance is a goal also for that demonstration since it allows for quick feedback on model success.

When it comes to scaling this up to larger models, examples of this are in the repository. We can try to give a better description of how this is done. It is for the most part about just having more equations, but the approach is the same as in a small model.

**There is also so much more could be done in such a framework and these dimensions are not considered, e.g. running the model multiple times, perturbing parameters to carry out sensitivity analyses, running ensembles of models, looking at model coupling in a more sophisticated way, and so on. Finally, there is**

**a lack of consideration of related work and yet there are a significant number of other frameworks in existence with similar goals.**

Running the model multiple times and sensitivity analysis is discussed around lines 60, 140, 175-195 (and demonstrated with source code in the Jupyter notebook in the link at line 199). Then one line 280-325 we do this for the specific study case. Note that MCMC is a form of sensitivity analysis that involves perturbation of parameters.

We also describe the API you can use to communicate with the model (exposed both in C++ and python and in principle any language with a C foreign function interface) (lines 136-140, and after 170). For instance we say that you can set new parameter values in the model, call the model run function, then read out the result values. It should be clear that this can be used for quickly scripting many kinds of sensitivity analysis setups (that are reusable across all Mobius models), or do other kinds of multiple model runs, set up ensemble runs, and to couple with external models.

We can make it more explicit in the final manuscript what is the possibilities and limitations of this API.

For instance you can also read and write input time series eg. in order to automate the setup of multiple climate scenarios (without being limited to using our specific file format).You can not have coupling with external models where there is a need for a per-timestep communication back and forth between the models (though this would not be that difficult to add if it is needed later) unless both models are created in Mobius. If you want to couple with an external model currently you need to take the entire model output of the other model as an input to the Mobius model (or the other way around). Apart from that, are there any other substantial forms of model coupling that should be addressed?

Other modelling frameworks are discussed in the paragraph starting at line 65. The point we make that we think separates our framework from others is that our models have a higher degree of customizability, while also not requiring the user to do much

explicit programming except for some very easy types of syntax. This may be clearer when we revise the description of the framework.

---

## Author Response (AR1)

**Author's response to reviews of "Rapid development of fast and flexible environmental models: The Mobius framework v1.0"**

We would like to thank the reviewers for their time and for the valuable comments!

**Reply to reviewer 1**

**R1: As a reader interested in general ode models paragraph 280 is not clear. If the goal is to compare manual vs autocalibration of model parameters the starting points should be independent (perhaps random in feasible ranges?). Why do the authors use the results of manual optimization to start the auto-calibration?**

AR: The goal was not to compare manual to autocalibration, but to show one possible workflow when calibrating the model and iterating on model structures (i.e. use autocalibration to "fine-tune" a manual calibration). We are not the authors of the autocalibration algorithms, and so the point is not to show off how good these are at reaching good solutions from random starting points. We are only showing one possible way these python libraries can be used with models built in Mobius. Feasibility of autocalibration would also be highly dependent on how complex the model structure is. This could be a selling point for a given model (it is easy to auto-calibrate), but the model we present is just there to illustrate how one can use the framework to iterate on and select a model structure, and this model itself is not the main point of the paper.

**R1: If the authors foresee that the changes in code would break the experiments and files referenced in the paper they can include the commit number (or date) that preserves the experiments since the repository seems to be actively developed.**

AR: Future edits to the github repository should not break the experiments described in the paper. However, we also made an archived version of the repository corresponding to this paper, which is linked to in section 5 (line 474).

**R1: The benchmark of runtime experiments can use a better description. Paragraph 335 mentions: "Results of the benchmarking show that Mobius models have a slight performance loss compared to hard-coded C++ models but run several orders of magnitude faster than hard-coded Python models (Table 3)". If the source files for these benchmarks are also included in the github repository, please reference them in the paper. The authors should give a simple description of the hardware they are using to run tests. Just the manufacturer, number of cores, and frequency of the CPU is enough. This will ensure the timings have enough context.**

AR: The source files for the benchmarking are referenced in the paper, line 403 "*Code used in these experiments can be found in the "Evaluation" subfolder of the Mobius repository*". Description of two of the machines used in benchmarking is given on line 408 "*for instance an Intel Core i5-6600K CPU 3.50 GHz, and an Intel Core i5-8350U CPU 1.70GHz*".

Note that we only report the ratio between the run times of different implementations. The run times themselves, are highly hardware dependent, while the ratios are fairly stable across several different machines we tried. What we want to show is how fast a framework-implemented model

runs compared to other implementation options. The number of cores is not going to be that relevant since a single model run is single threaded (while you can parallelize if you want to run the model many times such as in MCMC).

**R1: The timing experiments are average model evaluation runs. If possible (at least for the Mobius model using the python interface), It would be valuable to report optimization times in a separate table.**

AR: I'm not sure what precisely is asked for here. Optimization times are highly variable, depending on algorithm, the model used, input data etc. These together determine how many model evaluations are needed to reach convergence. The number of evaluations to reach convergence are independent of the model implementation (as long as the implementations give identical results). The model implementation only determines how fast each model evaluation is, and so that is what we focused on in the benchmarking. In other words the time for the optimization is roughly N*t, where N is the number of evaluations and t is the time for a single evaluation (the total time can be reduced if the algorithm allows parallelization, and t can be variable due to different convergence speed of ODE solvers depending on parameter values). The number N is not controlled by the model implementation, but by the optimization algorithm (and also the specific use case). Since the concern of this paper is the model implementation (framework), we focus on the single-run time in the benchmarking. Nevertheless, we did provide the timing for one particular optimization run on line 354 (using one specific algorithm, model setup and machine) just to give a general idea that using autocalibration is feasible in fast-iteration model building.

**Reply to reviewer 2**

**R2: This paper presents a framework to support the development of environmental models, referred to as the Mobius framework (v1.0). The aim is to allow scientists with potentially limited programming skills to develop component models within the framework, which can then be linked together. This is an important and timely contribution as new approaches to modelling are urgently required particularly as we head towards the need for integration of models. The framework is developed for hydrology and water quality analyses but should be applicable to other settings. The framework is also available as an open source tool with a link provided in the paper to the relevant github This is a well-written and accessible paper. My own major concern is how it is framed. It is written very much as a description of the approach rather than as a research paper. To be a fully-fledged research paper it would have: research questions and/ or a guiding hypothesis, consideration of the state of the art and gap identified, methodology and evaluation/reflections/discussion. These elements are largely missing. The paper would be much stronger being re-framed as more of a research contribution. I pick up on these points in my more specific comments below.**

AR: The main criticism is that the paper is not framed as a research paper. Please note that the paper was submitted as a 'Model description paper', which is a form outlined by GMD, and allows for descriptions of models and modelling frameworks. See https://www. geoscientific-model-development.net/about/manuscript_types.html Substantially evaluating if using a (or this) framework vs not using one is productive would make this into a different kind of paper, but that was not our intended scope.

**R2: The second section provides an overview of Mobius. I found it quite hard to get to the crux of the design, and it is quite short and lacks any real depth. I am a computer scientist by training and I wanted to see things like an overarching software architecture and also an explanation of key design decisions with rationale. This is missing from me. It would be very hard fro example for other researchers to consider the text here and get anywhere near reproducing 'the approach'.**

AR: Note that the form outlined by GMD says that implementation details should be included if they will substantially affect the output results. For models this is straightforward, but for frameworks what this implies is a little more up to interpretation. Note also that the documentation included in the repository (and the source code itself) is an important part of the supplementary material to the paper, which does go into more detail. See especially https://github.com/NIVANorge/Mobius/blob/master/Documentation/ model_builder_documentation.pdf for a detailed description of how one uses the framework to build models.

We have made a major revision of section 2 of the paper, which now describes the software architecture in more detail and gives a glimpse into the implementation. We decided to not go into more details about implementation since this is not meant to be a CS paper about programming techniques, but instead is targeted to environmental scientists who may want to use a framework like Mobius rather than reimplement it. The source code and documentation can be further used by researchers who want to reproduce the techniques.

**R2: Section 3 is then a 'demonstration of Mobius' and this title says a lot about the way the paper is framed. To me, it should not be a demonstration of a given approach but rather should be an evaluation of how well the approach achieves its goals, with the evaluation being rigorous and scientific. Instead, it steps through the GUI (but curiously not in a visual way) and also the use of Python wrappers (the key to interoperability in their approach), but not in a way that allows the reader to see beyond the "what" to the "why" this is done (and other alternatives that could have been considered). This section also contains a case study – but again its stated purpose is to demonstrate not to evaluate. It is also quite a small example and it is not clear how this would scale up to something more substantial. The section concludes with some benchmark figures, which are interesting, but it is not clear why performance is measured and nothing else is evaluated, when performance is not mentioned as a goal. Section 4 contains a discussion but to me this is way to narrow and specific and lacks a true element of reflection, e.g. what has worked, what has not worked, what are the strengths of the approach and weaknesses, and so on.**

AR: As we pointed out above, the reason why the paper is framed the way it is, is that it is intentionally a description paper and not a research paper.

We have added many paragraphs to the discussion section (section 4) that has more reflection on what has worked and what hasn't.

There is a screen shot of the GUI (figure 1, line 550), which shows a substantial part of its functionality. We have added some more description to the figure caption. (note that in the submission form of the manuscript all figures are at the end of the document. Presumably they will be inserted at appropriate places during editing). The GUI is not the main focus of the paper, but is described briefly because of how it aids in making the framework easy to use both during development and for third-party users of the finished models.

The purpose of the demonstration section (section 3) is to show that the framework allows you to build, modify and evaluate models, with the goals that are stated earlier in the paper, among others:

1. it is very little work to modify models in order to evaluate different model structures.

2. It is easy to reuse existing modules (in this case a hydrology module)

3. Speed of model runs (performance) (this is definitely one of the important stated goals, see e.g. the paragraph around line 60)

4. Models can easily be run many times such as in auto-optimization and sensitivity analysis, which are important tools for evaluating model success and identify structural issues in the models. This is stated as a goal around line 60:

*"Although modelling frameworks have many advantages compared to traditional (fixed structure) approaches, they are difficult to implement in practice as building a flexible, generalizable framework requires considerably more programming expertise than building a static model. Many modellers have skills in interpreted languages such as Python or R, which are reasonably well-suited to rapid prototyping of alternative model structures. However, exploring uncertainty in model predictions, or formally comparing model structures, typically requires methods such as Bayesian MCMC, which involves running each model thousands or even millions of times. This can be a big limitation for models coded in interpreted languages, which are comparatively slow. One solution is to build models using compiled languages like C++ or Fortran, but many modellers lack the time or inclination to develop the necessary programming skills."*

It ties in to why it is important that the models run fast. We show that having auto-optimization and sensitiviy algoritms that can be used with any model built in the framework is useful because it allows you to very quickly apply them to any newly created model structures with little extra effort.

The statistics given in Table 2 (line 562) do show rigorously that autocalibration helps you to arrive at good calibrations (in a specific example) and that it informs you when selecting model structures. Figure 3 (line 556) along with the paragraph starting at line 385 shows rigorously how MCMC can aid in identifying structural issues in models, like autocorrelation and sensitivity of parameters.

To see if the claims about ease of use hold one can also inspect the source code used in the examples, which is given as a part of the supplementary material.

We don't say that using python is the key to interoperability, but we claim that it is one useful way to do it since it tends to be accessible to domain scientists and has many sensitivity+uncertainty analysis, optimization and other useful packages readily available. Section 2.2 describes to a greater extent the API that allows for interoperability (you can use other languages than python).

When it comes to scaling this up to larger models, examples of this are in the repository. For the most part it amounts to just having more model entities, but the approach is the same as in a small model. This is also now discussed in the discussion section.

**R2: There is also so much more could be done in such a framework and these dimensions are not considered, e.g. running the model multiple times, perturbing parameters to carry out sensitivity analyses, running ensembles of models, looking at model coupling in a more sophisticated way, and so on. Finally, there is a lack of consideration of related work and yet there are a significant number of other frameworks in existence with similar goals.**

AR: Running the model multiple times and sensitivity analysis is discussed around lines 60, 197-200 (and demonstrated with source code in the Jupyter notebook in the link at line 260). Then on line 385-395 we do this for the specific study case. Note that MCMC is a form of uncertainty and sensitivity analysis that involves perturbation of parameters. In section 2.2 we describe the API you

can use to communicate with the model (exposed both in C++ and python and in principle any language with a C foreign function interface). For instance we say that you can set new parameter values (or input data) in the model dataset, call the model run function, and read out the result values. This can easily be used to script specific sensitivity analysis setups, other types of multiple model runs etc.

Coupling of models is now discussed some more on line 456.

Other modelling frameworks are discussed in the paragraph starting at line 65:

*"Several hydrological model building frameworks exist to date, such as FUSE (Clark et al., 2008), SUPERFLEX (Fenicia et al., 2011), FARM (Euser et al., 2013) and SUMMA (Clark et al., 2015). These all allow predefined components to be connected in user-specified ways to create a model, with a focus on catchment hydrology. The framework presented here takes these existing approaches further, allowing the user to define any component/process. It is therefore, to our knowledge, one of the first frameworks to be fully generalizable: although initially developed to support catchment-scale hydrology and water quality modelling, it can be used to represent any system of hierarchically structured ordinary differential equations (ODEs), such as population dynamics or toxicological models. Moreover, the Mobius framework does more of the heavy lifting of organizing the program structure than what we find in other frameworks. The user can create models with a complex structure without having to organize that structure explicitly using programming architecture."*

Why these claims hold should be clearer from the new version of section 2. See also the paragraph:

*"One benefit of the fact that the framework organizes the main evaluation loops of the model is that ODE equations from a module can be solved in the same ODE integrator batch as the ODE equations from another module without any effort by the model creator. In this way one module can extend the ODE systems of another, creating a larger coupled set of equations, or even override individual equations from the other module without creating a separate version of the source code of that other module."*

Some frameworks allow users to extend coupled ODE systems. None that we know of allow replacing individual equations from other modules, defining new ODE systems (choosing what integrator algorithm to use), have a completely flexible set of index sets that evaluation should loop over (differently for different batches of equations). The Mobius framework lets the user specify such things (with a few simple function calls) while not having to do the heavy programming that enables it. This is described several places in the paper.